# ON-TARGET ADAPTATION

## ABSTRACT

Domain adaptation seeks to mitigate the shift between training on the *source* data and testing on the *target* data. Most adaptation methods rely on the source data by joint optimization over source and target. Source-free methods replace the source data with source parameters by fine-tuning the model on target. Either way, the majority of the parameter updates for the model representation and the classifier are derived from the source, and not the target. However, target accuracy is the goal, and so we argue for optimizing as much as possible on target. We show significant improvement by *on-target adaptation*, which learns the representation purely on target data, with only source predictions for supervision (without source data or parameter fine-tuning). In the long-tailed classification setting, we demonstrate on-target class distribution learning, which learns the (im)balance of classes on target data. On-target adaptation achieves state-of-the-art accuracy and computational efficiency on VisDA-C and ImageNet-Sketch. Learning more on target can deliver better models for target.

## 1 INTRODUCTION

Deep networks achieve tremendous success on various visual tasks at the expense of massive data collection and annotation efforts. Even more data is needed when training (source) and testing (target) data differ, as the model must be adapted on the new data to maintain accuracy. To reduce the annotation effort on new data, unsupervised domain adaptation (UDA) approaches transfer knowledge from labeled source data to unlabeled target data. Standard UDA requires simultaneous optimization on the source and target data to do so. However, this requirement may not be entirely practical, in that *shifted* or *future* target data may not be available during training. Furthermore, (re-)processing source data during testing may be limited by computation, bandwidth, and privacy. Most importantly, it is the target data that ultimately matters for testing. In this work, we therefore turn our attention from source to target, and how to learn more from it.

Recent work adapts to the target data without the source data or even adapts during testing. However, these "source-free" and "test-time" approaches still rely heavily on the source parameters for fine-tuning. Source-free adaptation initializes from source parameters then optimizes on target data without the joint use of source data (Liang et al., 2020; Li et al., 2020; Kundu et al., 2020). Test-time adaptation partially updates source parameters on the target data while testing (Sun et al., 2019; Schneider et al., 2020; Wang et al., 2021). Such approaches reduce reliance on the source data, and can even improve accuracy, but have they made full use of the target data? Many of the model parameters are fixed (Liang et al., 2020; Schneider et al., 2020; Wang et al., 2021) or regularized toward the source parameters (Li et al., 2020; Kundu et al., 2020). We investigate whether more can be learned from target, and more accuracy gained, by not transferring the source parameters.

We propose on-target adaptation to unshackle the target representation from the source representation. To do so, we (1) factorize the representation from the classifier and (2) separate the source parameters from the source predictions. By factorizing the representation from the classifier, we can train the representation entirely on the target data by self-supervision. Given this on-target representation, we can then supervise a new classifier from source predictions by distillation (Hinton et al., 2015), without transferring the source parameters. Not transferring parameters frees our target model from the constraints of the source architecture, so that we can experiment with distinct target architectures. In this way, we can even change the model size to optimize a target-specific model that is more accurate and more efficient. In contrast to prior work on adaptation, this uniquely allows for learning $100\%$ of the target model parameters on target data, as illustrated by Figure 1.

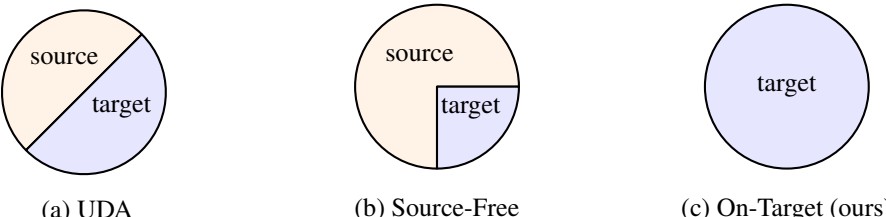

(a) UDA      (b) Source-Free      (c) On-Target (ours)

Figure 1: Domain adaptation adjusts a model trained on source data for testing on target data. We contrast methods by their updates on source and target. Unsupervised domain adaptation (UDA) jointly learns $50/50$ on source/target. Source-free adaptation transfers source parameters, then selectively learns on target. Our *on-target* approach learns $100\%$ of the testing model parameters on target by neither sharing nor transferring source parameters, but instead distilling source predictions.

To realize our proposed factorization and separation, we employ contrastive learning, source-free adaptation, and teacher-student distillation. We initialize the target representation by self-supervision with contrastive learning. We turn the source model into a teacher model by source-free adaptation, and then generate pseudo-labels to supervise distillation. We lastly train the student model on the teacher supervision, starting from the target representation and new classifier parameters, and repeat this teacher-student cycle by resetting the student classifier parameters between epochs. Contrastive learning has recently enabled self-supervised representations to compete with or even surpass supervised representations (Chen & He, 2020; Caron et al., 2020; He et al., 2020; Chen et al., 2020; Grill et al., 2020; Zbontar et al., 2021). We show it provides a sufficient target representation.

Our experiments show on-target adaptation achieves state-of-the-art accuracy and computational efficiency on common domain adaptation benchmarks. For model accuracy, our method brings $\sim 3\%$ absolute improvement compared to state-of-the-art unsupervised and source-free domain adaptation methods on VisDA-C (Peng et al., 2017) and ImageNet Sketch (Wang et al., 2019a) while reducing $50\%+$ of parameters. For computation, our method reduces FLOPs by $50+\%$ and memory by $75+\%$ for each forward pass of the target model. In the long-tailed classification setting, on-target class distribution learning equals the state-of-the-art learnable weight scaling (Kang et al., 2019) without needing source data. Ablation experiments support the generality of on-target representation learning across architectures, contrastive learning methods, losses, and amount of optimization.

Our contribution is to investigate whether the source data should be the primary source of target model parameters, and to propose an alternative: on-target adaptation. Our insight is that the source representation can be fully decoupled from source supervision. Domain adaptation normally emphasizes the representation of source data, by either jointly optimizing on source data or transferring source parameters. On-target adaptation emphasizes the representation of target data instead, by distilling source predictions into a self-supervised target representation. We are the first to show this is feasible, as a new kind of source-free adaptation. Furthermore we show it improves accuracy and reduces computation on standard benchmarks like VisDA-C.

## 2 RELATED WORK

**Adaptation** On-target adaptation is unique in its decoupling of the target representation from the source representation. Prior adaptation approaches transfer the source representation to the target, either by joint optimization or by initialization. To transfer the source model to a visually different target domain, unsupervised domain adaptation (UDA) learns a joint representation for both domains for visual recognition tasks, such as image classification (Tzeng et al., 2014), object detection (Chen et al., 2018), semantic segmentation (Hoffman et al., 2016). Some of the most representative unsupervised domain adaptation ideas are 1) maximum mean discrepancy (Long et al., 2015; 2017); 2) moment/correlation matching (Sun et al., 2016; Zellinger et al., 2017); 3) domain confusion (Ganin & Lempitsky, 2015; Tzeng et al., 2017); 4) GAN-based alignment (Liu et al., 2017; Hoffman et al., 2018). All these UDA methods need simultaneous access to both source and target data. In practice, it might be impossible to meet this requirement due to limited bandwidth, computational power, or privacy concerns. Therefore, test-time training (Sun et al., 2019), source-free adaptation (Liang et al., 2020), and fully test-time adaptation (Wang et al., 2021) settings focus on adapting a source

Figure 2: On-target adaptation proceeds in four stages. Source data is colored in orange while target data is colored in blue. Stage 0 is the only stage to use source data. Stage 3 is the stage that connects source and target: the target representation from stage 2 is fine-tuned as a student model on the predictions of the teacher model from stage 1. Note that no parameters are shared or transferred from source to target, so the target parameters are fully learned on target data.

model by fine-tuning on the target data without source data. Exciting concurrent work even adapts without the source model by only using source predictions (Liang et al., 2021; Zhang et al., 2021; Wu et al., 2021). These "black-box" adaptation methods exclusively optimize teacher and student predictions, with distillation losses and output regularizers. While we likewise apply teacher-student learning, our work is complementary in using contrastive learning as a loss on the input for the student representation.

**Semi-supervised learning** Many UDA methods follow the practice of semi-supervised learning, especially pseudo labeling (Lee, 2013) which is to utilize the model prediction to generate supervision for the unlabeled images. The typical setup of unsupervised domain adaptation methods is to jointly optimize with ground truths on the source and pseudo labels on the target (Zhang et al., 2018; Choi et al., 2019; Long et al., 2017; Zou et al., 2018). When source data annotations are not available, DeepCluster (Caron et al., 2018) and SHOT (Liang et al., 2020) further leverage weighted k-means clustering to reduce the side effects on noisy pseudo labels. Similarly, our method does not require access to labeled source data, while only relying on the target images with generated pseudo labels. In addition, our method heavily benefits from the contrastive learned target domain representation, which is treated as initialization to overcome the misleading of noisy pseudo labels.

**Long-tailed recognition** Long-tailed recognition tackles imbalanced class distributions in real-world data. Existing work divides into three groups: 1) re-balancing the data distribution (Chawla et al., 2002; Han et al., 2005; Shen et al., 2016; Mahajan et al., 2018); 2) designing class-balanced losses (Cui et al., 2019; Khan et al., 2017; Cao et al., 2019; Khan et al., 2019; Huang et al., 2019; Lin et al., 2017; Shu et al., 2019; Ren et al., 2018; Hayat et al., 2019); 3) transfer learning across classes (Yin et al., 2019; Liu et al., 2019). All of these methods address imbalance by altering training, so that the model may learn more balanced features, and a classifier that covers common (head) and rare (tail) classes. We instead adapt the classifier for long-tailed recognition during testing.

## 3 METHOD: ON-TARGET ADAPTATION

The goal of the proposed on-target adaptation is to tackle domain shift during the test-time with only a source model, without the access of annotation and source data. Specifically, the supervised model with source parameter $f(\cdot; \theta^s)$ trained on source images $x^s$ and labels $y^s$ needs to generalize on unlabeled target data $x^t$ when an unneglectable domain shift happened. Our on-target adaptation (Figure 2) is proposed to obtain target model parameter $\theta^t$ purely during test-time.

**Stage 0 (source): train model with labeled source data** We train a deep ConvNet and learn source parameter $\theta^s$ by minimizing vanilla cross-entropy loss $\mathcal{L}(\hat{y}^s, y^s)$ on labeled source data $(x^s, y^s)$. Specifically, $\mathcal{L}(\hat{y}^s, y^s) = -\Sigma_c p(y_c^s) \log(p(\hat{y}_c^s))$ for the predicted probability $\hat{y}_c^s$ of class $c$, where target probability $y_{gt}^s$ is 1 for the ground truth class $gt$ and 0 for the rest.

**Stage 1 (teacher): adapt source model without source data** We update the source parameter $\theta^s$ during testing to minimize information maximization (InfoMax) loss (Gomes et al., 2010). Specifically, InfoMax loss augment entropy loss $\mathcal{L}_{ent} = -\Sigma_c p(\hat{y}_c^t) \log(p(\hat{y}_c^t))$ with diversity objective $\mathcal{L}_{div} = D_{KL}(\hat{y}^t \parallel \frac{1}{C}\mathbf{1}_C) - \log(C)$. where $D_{KL}$ indicates the KullbackLeibler divergence, $\mathbf{1}_C$ is an all-one vector with $C$ dimensions. Here $\frac{1}{C}\mathbf{1}_C$ indicates the target label vector with evenly distributed $\frac{1}{C}$ probabilities, where $\mathcal{L}_{div}$ is propose to enforce the global diversity over classes.

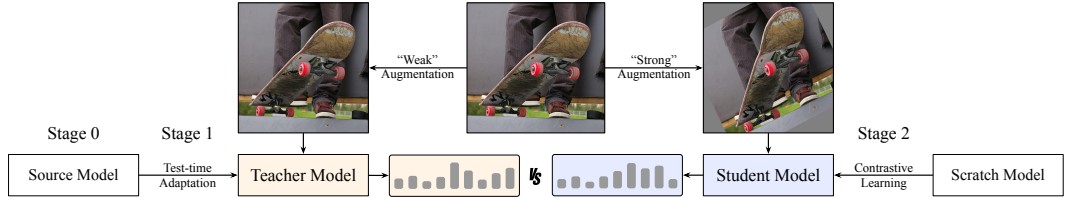

Figure 3: Teacher-student (stage 3) learning in our method. Transfer learning between the teacher (orange) and the student (blue), where pseudo labels are generated on the weakly-augmented images. The model is trained on the strongly-augmented target data to match the pseudo labels.

As for the parameters to optimize over, we follow the motivation of decoupling the representation and classifier. When the classifier is frozen, the goal of optimization is to mitigate domain shift by deriving proper target features from the source model. In particular, we keep the classifier the same on both source and target domain, and obtain $\Delta$ by the gradient of the test-time objective (InfoMax), to update the representation part of model parameter $\theta^s$.

**Stage 2 (student): initialize target model with contrastive learning** Instead of fine-tuning from source model, we choose to initialize the target feature purely from target data. Benefiting from the recent advances in contrastive learning methods, we train an unsupervised model with purely unlabeled target images. Specifically, we initialized target representation via improved momentum contrast learning (MoCo v2) (He et al., 2020; Chen et al., 2020). It is worth noting that our method does not require a specific contrastive learning method. In other words, the default MoCo v2 could be easily replaced by a more recent self-supervised learning model, such as SwAV (Caron et al., 2020), SimSiam (Chen & He, 2020), Barlow Twins (Zbontar et al., 2021). Such a modular design makes it easier to benefit from the latest advance in contrastive learning. We have performed an ablation study on the choice of contrastive learning method in Section 4.5.

**Stage 3 (teacher-student): transfer knowledge from teacher to student** We use the adapted source model $f(\cdot; \theta^s + \Delta)$ as the initial teacher model to generate pseudo labels $y'^t$ on unannotated target images $x^t$. Then we fine-tune the student model $f(\cdot; \theta^t)$ initialized by contrastive learning on target data with cross-entropy loss $\mathcal{L}(\hat{y}^t, y'^t) = -\Sigma_c p(y_c'^t) \log(p(\hat{y}_c^t))$. Specifically, we use normal distribution with a mean of zero and standard deviation of 0.01 for the classification head, since contrastive learned model does not contain classifier. The teacher would be replaced with the latest student to gradually denoise pseudo labels for the subsequent phase. Meanwhile, the contrastive learned model would re-initialize the student feature to eliminate the accumulated errors from imperfect pseudo labels. In other words, the student model would start over one more time with only newer pseudo labels for the next transferring phase.

Figure 3 illustrates the procedure of transferring the knowledge from teacher to student. Specifically, the interaction between teacher and student models benefits from consistency regularization and pseudo-labeling, inspired by a recent semi-supervised learning approach called FixMatch (Sohn et al., 2020). During the transferring, we augment the target images with random cropping, random flipping, and AutoAugment with ImageNet policy, as "strong" augmentation, while the "weak" augmentation is the combination of resizing and center cropping when generating pseudo labels. Relying on the assumption that the model should generate similar predictions on data-augmented versions of the same image (Bachman et al., 2014; Sajjadi et al., 2016; Laine & Aila, 2016), consistency regularization enforces the cross-entropy loss between student output on strongly-augmented images and teacher output on weakly-augmented images.

# 4 EXPERIMENTS

## 4.1 SETUP

**Datasets** We evaluate our method on both domain adaptation and long-tailed recognition benchmarks, including VisDA-C (Peng et al., 2017), Office Home (Venkateswara et al., 2017), Sketch (Wang et al., 2019a), ImageNet-LT (Liu et al., 2019), and iNaturalist-18 (Van Horn et al., 2018). Figure 5 presents some of example images to illustrate domain shifts.

**Metric** We report top-1 accuracy (denoted as acc.) on the whole dataset for all datasets. On VisDA-C, we additionally report the percentage accuracy of each category and the corresponding average of all categorical accuracies (denoted as avg.), due to the imbalanced distributed label space. On Office Home, we calculate the average for all kinds of domain shifts as the summary number for each method. On long-tailed recognition benchmarks, we additionally report percentage accuracy on many-shot (more than 100 samples), medium-shot (20-100 samples), and few-shot (less than 20 samples) following the evaluation protocol from Liu et al. (2019); Kang et al. (2019).

**Baselines** We choose the most recent fully test-time adaptation method, TENT (Wang et al., 2021), and source-free adaptation framework, SHOT (Liang et al., 2020), as our "online" and "offline" adaptation baselines. TENT does not alter training, while SHOT minimally customizes the source architecture and training. Entropy minimization is the target optimization objective for both TENT and SHOT. SHOT additionally regularizes optimization by the information maximization (InfoMax) loss (Gomes et al., 2010; Shi & Sha, 2012; Hu et al., 2017) to augment entropy minimization on each sample with diversity maximization across samples. Following SHOT, we therefore augment TENT into TENT-IM by including this regularization. Alongside their role as baselines, these methods can serve as teacher models for our stage 1. We also compare with unsupervised domain adaptation (UDA) baselines, including DANN (Ganin & Lempitsky, 2015), DAN (Long et al., 2015), ADR (Saito et al., 2018), CDAN+E (Long et al., 2018), CDAN+BSP (Chen et al., 2019), CDAN+TN (Wang et al., 2019b), SAFN (Xu et al., 2019), SWD (Lee et al., 2019), DSBN+MSTN (Chang et al., 2019), STAR (Lu et al., 2020). It is worth noting that all these UDA methods are fine-tuning from the ImageNet pretrained ResNet-101 source model, with access to both source and target data. TENT-IM and SHOT likewise initialize their representation from the source model. In contrast, our method trains ResNet-18 from scratch, and entirely on the target data, by contrastive learning for the representation and teacher-student learning for the classification.

For long-tailed recognition, we choose learnable weight scaling (LWS) (Kang et al., 2019) as the train-time baseline. LWS first decouples the full model into representation and classification, and then only re-scales the classifier parameters with class-balanced sampling. Our on-target class distribution learning extends LWS to test-time, by re-scaling the parameters on unlabeled target data.

**Architecture** For comparability with state-of-the-art models, we choose 18/50/101-layer ResNet models (He et al., 2016) for both main results and ablation studies. When reproducing the prior works, we keep the architecture the same, for example, the weight-normalization (Salimans & Kingma, 2016) augmented ImageNet pretrained ResNet-101 for SHOT (Liang et al., 2020).

## 4.2 IMPLEMENTATION

Our implementation is in PyTorch (Paszke et al., 2019) and depends on the VISSL (Goyal et al., 2021), MMClassification (Contributors, 2020), and Weights & Biases (Biewald, 2020) libraries. The code is attached and will be released for publication.

**Stage 0 (source)** We train residual networks (He et al., 2016) with various depths (including 18, 50, 101), and initializations (ImageNet pretraining or Kaiming init (He et al., 2015) when training from scratch). We optimize cross-entropy loss by SGD with an initial learning rate 0.1, momentum 0.9, weight decay 0.0001, batch size 256. We do not apply label smoothing (Müller et al., 2019) except for SHOT (Liang et al., 2020), as it specifically includes it. We adopt the standard data augmentation pipeline from ImageNet training, such as random cropping, random flipping, and color jitter. We choose ImageNet statistics as the default input mean and variance for all models.

**Stage 1 (teacher)** We experiment with three types of teacher models: 1) source-only, 2) TENT-IM (Wang et al., 2021), 3) SHOT (Liang et al., 2020). To optimize this altered loss, we choose SGD with learning rate 0.0001, momentum 0.9, and weight decay 0.0001. In addition to batch normalization (Ioffe & Szegedy, 2015), we also update convolutional layers except the final classification layer. As for SHOT, We execute the authors' open-sourced codebase with the same hyper-parameters for various architectures (ResNet-50 and ResNet-101), initialization (from scratch and ImageNet pretrain), and domain shifts (train to val/test splits on VisDA-C).

**Stage 2 (student)** We experiment with two designs as students: 1) source-only, 2) contrastive learning. Specifically, we leverage some of off-the-shelf contrastive learning methods to initialize target-domain representation, such as MoCo v2 (Chen et al., 2020), SimSiam (Chen & He, 2020),

| method | #1 source-free adaptation | #2 contrastive learning | #3 teacher student | VisDA-C train →val | →test |
|---|---|---|---|---|---|
| source-only | | | | 21.8 | 23.9 |
| source-free | ✓ | | | 31.4 | 34.3 |
| iterated distillation | | | ✓ | 28.3 | 31.8 |
| with adaptation | ✓ | | ✓ | 43.3 | 46.0 |
| on-target adaptation | | ✓ | ✓ | 29.1 | 33.9 |
| | ✓ | ✓ | ✓ | 49.9 | 51.2 |

Table 1: Each stage of our on-target adaptation improves target accuracy. Contrastive learning (stage 2), for fitting the representation on target data alone, helps whether or not the teacher is adapted (stage 1).

**Require:** model $f_s$ # source model, for predictions
**Require:** model $f_t$ # target model, for representation
**Require:** image transform $t_w$ # weak augmentation
**Require:** image transform $t_s$ # strong augmentation
$f_0 \leftarrow f_s$ # source model for the first teacher
**for** i ← 1 to N **do** # for each epoch
   $f_i \leftarrow f_t$ # initialize by contrastive learning
   **for** b ← 1 to B **do** # for each mini-batch
      $y' \leftarrow f_{i-1}(t_w(x))$ # teacher
      $\hat{y} \leftarrow f_i(t_s(x))$ # student
      $\ell \leftarrow \text{Loss}(\hat{y}, y')$ # hard-label cross-entropy
   **end for**
**end for**
**return** $f_N$ # student model for testing

Figure 4: On-target pseudo-code.

| method | network | plane | bcycl | bus | car | horse | knife | mcycl | person | plant | sktbrd | train | trunk | avg. | acc. |
|---|---|---|---|---|---|---|---|---|---|---|---|---|---|---|---|
| ADR (Saito et al., 2018) | R101P | 94.2 | 48.5 | 84.0 | 72.9 | 90.1 | 74.9 | 92.6 | 72.5 | 80.8 | 61.8 | 82.2 | 28.8 | 73.6 | 73.8 |
| CDAN+E (Long et al., 2018) | R101P | 85.2 | 66.9 | 83.0 | 50.8 | 84.2 | 74.9 | 88.1 | 74.5 | 83.4 | 76.0 | 81.9 | 38.0 | 73.9 | 71.0 |
| CDAN+BSP (Chen et al., 2019) | R101P | 92.4 | 61.0 | 81.0 | 57.5 | 89.0 | 80.6 | 90.1 | 77.0 | 84.2 | 77.9 | 82.1 | 38.4 | 75.9 | 73.4 |
| SAFN (Xu et al., 2019) | R101P | 93.6 | 61.3 | 84.1 | 70.6 | 94.1 | 79.0 | 91.8 | 79.6 | 89.9 | 55.6 | 89.0 | 24.4 | 76.1 | 75.6 |
| SWD (Lee et al., 2019) | R101P | 90.8 | 82.5 | 81.7 | 70.5 | 91.7 | 69.5 | 86.3 | 77.5 | 87.4 | 63.6 | 85.6 | 29.2 | 76.4 | 75.5 |
| DSBN+MSTN (Chang et al., 2019) | R101P | 94.7 | 86.7 | 76.0 | 72.0 | 95.2 | 75.1 | 87.9 | 81.3 | 91.1 | 68.9 | 88.3 | 45.5 | 80.2 | 79.2 |
| STAR (Lu et al., 2020) | R101P | 95.0 | 84.0 | 84.1 | 73.0 | 91.6 | 91.8 | 85.9 | 78.4 | 94.4 | 84.7 | 87.0 | 42.2 | 82.7 | 80.4 |
| TENT-IM (Wang et al., 2021) | R50S | 58.9 | 40.1 | 50.2 | 23.6 | 22.6 | 25.3 | 29.8 | 24.8 | 22.9 | 30.2 | 45.1 | 20.1 | 32.8 | 31.4 |
| TENT-IM + Ours | R50S | 90.1 | 66.0 | 75.2 | 41.3 | 29.2 | 11.2 | 57.0 | 60.8 | 40.1 | 51.1 | 73.6 | 23.8 | 51.6 | 50.9 |
| TENT-IM + Ours | R18S | 90.5 | 65.2 | 79.6 | 38.8 | 26.7 | 12.9 | 51.6 | 59.9 | 44.2 | 46.0 | 71.1 | 24.0 | 50.9 | 49.9 |
| SHOT (Liang et al., 2020) | R101P | 94.6 | 86.6 | 79.5 | 55.6 | 93.6 | 96.1 | 79.8 | 80.7 | 89.2 | 89.0 | 86.1 | 57.1 | 82.3 | 77.8 |
| SHOT + Ours | R18S | 96.0 | 89.5 | 84.3 | 67.2 | 95.9 | 94.2 | 91.0 | 81.5 | 93.8 | 89.9 | 89.1 | 58.2 | 85.9 | 82.8 |

Table 2: Classification accuracy of on-target adaptation on VisDA-C (validation) across all categories and averaged over classes (avg.) and images (acc.). R18/50/101S denotes ResNet-18/50/101 randomly initialized from scratch and R18/50/101P denotes ResNet-18/50/101 pretrained on ImageNet.

SwAV (Caron et al., 2020), Barlow Twins (Zbontar et al., 2021). Compared to their training recipes on ImageNet, we have more epochs on VisDA-C val/test with the same batch size, learning rate, data augmentation, and model architecture, to make the training procedure longer with the smaller amount of images.

**Stage 3 (teacher-student)** By default, the whole knowledge distillation consists of three phases, where each phase has 10 epochs to train the student model with the hard pseudo label. The student would be reset to the contrastive model to avoid error accumulation at the beginning of every phase. The teacher would be replaced with the latest student before starting the next phase, so that the quality of pseudo-labeling could be improved gradually. We utilize SGD with an initial learning rate 0.01, momentum 0.9, weight decay 0.0001, batch size 256, and cosine annealing scheduler (Loshchilov & Hutter, 2016).

## 4.3 On-target adaptation

Table 1 reports reports the change in target accuracy for each stage of on-target adaptation on VisDA-C. The source model is ResNet-50 trained from scratch. The target model is ResNet-18. Source-free adaptation is done by TENT-IM. In teacher-student learning, the student, with either the source (teacher) or our on-target (contrastive) representation, is trained on the predictions of the teacher. This is repeated for multiple epochs. The on-target representation (stage #2) improves accuracy with and without source-free adaptation of the teacher, as it only depends on the target data.

**VisDA train → val** Table 2 compares our method with state-of-the-art unsupervised (upper part) and test-time (lower part) domain adaptation approaches from VisDA-C train to val splits. The proposed on-target adaptation significantly improves the existing test-time adaptation methods. It is worth noting that all these existing methods need to keep the same architecture when joint-training or fine-tuning on the source model. On the contrary, our method could utilize a much more lightweight model, such as ResNet-18 as shown in this table. For example, our method brings 18+ points improvement compared to source-free adapted teacher TENT-IM, while reducing over 50% parameters and runtime flops, and 75% memory consumption at each feed-forward of target model.

**VisDA train → test** Table 3 compares our method with state-of-the-art unsupervised (upper part) and test-time (lower part) domain adaptation approaches from VisDA-C train to test splits. Similar to table 2, our method dramatically improves the performance of all kinds of teacher models.

| | source-only | | ours | | | | TENT-IM | | ours | | | | SHOT | | ours | |
|---|---|---|---|---|---|---|---|---|---|---|---|---|---|---|---|---|
| network | avg. | acc. | avg. | acc. | | network | avg. | acc. | avg. | acc. | | network | avg. | acc. | avg. | acc. |
| R50S | 22.1 | 23.9 | 30.9 | 33.9 | | R50S | 34.2 | 34.3 | 49.3 | 51.2 | | R101P | 89.3 | 88.4 | 91.7 | 91.6 |
| R50P | 34.4 | 37.7 | 39.8 | 43.5 | | R50P | 60.4 | 62.4 | 74.8 | 77.7 | | R50P | 76.4 | 78.0 | 81.1 | 83.0 |

Table 3: Classification accuracy of our method supervised by three teachers: source-only, SHOT, and TENT-IM on VisDA-C (test). R50/101S denotes ResNet-50/101 randomly initialized from scratch and R50/101P denotes ResNet-50/101 pretrained on ImageNet.

| method | network | accuracy |
|---|---|---|
| Anisotropic (Mishra et al., 2020) | ResNet-50 | 24.5 |
| Debiased (Li et al., 2021) | ResNet-50 | 28.4 |
| Crop (Hermann et al., 2020) | ResNet-50 | 30.9 |
| RVT (Mao et al., 2021) | DeiT-B | 36.0 |
| TENT-IM (Wang et al., 2021) | ResNet-50 | 35.6 |
| TENT-IM + Ours | ResNet-18 | 37.5 |
| TENT-IM + Ours | ResNet-50 | 40.5 |

Table 4: Adaptation on ImageNet-Sketch.

| method | network | accuracy |
|---|---|---|
| DANN (Ganin & Lempitsky, 2015) | ResNet-50 | 57.6 |
| DAN (Long et al., 2015) | ResNet-50 | 56.3 |
| CDAN+E (Long et al., 2018) | ResNet-50 | 65.8 |
| CDAN+BSP (Chen et al., 2019) | ResNet-50 | 66.3 |
| SAFN (Xu et al., 2019) | ResNet-50 | 67.3 |
| CDAN+TN (Wang et al., 2019b) | ResNet-50 | 67.6 |
| TENT-IM (Wang et al., 2021) | ResNet-50 | 53.8 |
| TENT-IM + Ours | ResNet-18 | 56.8 |

Table 5: Adaptation on Office-Home.

In the following two paragraphs, we discuss the potential application of on-target adaptation *without contrastive learning*. When test-time data is not sufficient enough to finish the contrastive learning, we could skip contrastive learning on target data (stage 2) of the proposed method. In other words, we directly fine-tune the target model initialized by the source model. We believe that adaptation performance could be further improved once the contrastive learning could no longer be data-hungry or target domain data could be abundant.

**ImageNet → Sketch** Table 4 reports the empirical results on generalization regarding ImageNet/Sketch as source/target domain. For on-target adaptation, we try two student models: model as same as the teacher model (ResNet-50) and small supervised model pretrained on ImageNet (ResNet-18). Our method additionally brings ∼5/3% improvements compared to the teacher model with the same/shallower student models, where the teacher model, TENT-IM, already outperforms the previous state-of-the-art by ∼5%.

**Office Home** Table 5 compares our method with state-of-the-art unsupervised (upper part) and test-time (lower part) domain adaptation approaches for the various domain shifts in Office Home. Our method advances the average accuracy over all domain shifts of teacher model (TENT-IM) by 3%.

### 4.4 ON-TARGET CLASS DISTRIBUTION LEARNING

In this section, we argue for calibrating classifier during the test-time, without any modification on training procedure, aiming at long-tailed recognition task. Here we treat the long-tailed data as the source domain while regarding the class-balanced data as the target domain. During training, instance-balanced sampling provides a generalizable representation to start with. Then the classifier is re-scaled during the test-time on the class-balanced data.

First, we train the source domain model with the instance-balanced sampling, which samples each sample with the same probability. In this way, the learned classifier has a higher prior probability on the head compared to the tail. Then we calibrate the parameters of the classifier while freezing the feature with test images and pseudo labels, following the practice of our on-target adaptation. It is worth mentioning that we do not utilize contrastive learning to re-initialize the representation on target data or tune the feature part in teacher-student (stage 3). The major reason for such a choice is to follow the practice of LWS (Kang et al., 2019), which points out that the domain shift only exists within class distribution so that only classifier needs to be calibrated.

The empirical results indicate that our method could automatically calibrate the categorical prior and adaptive fit the test data distribution without the access of training data. Table 6 demonstrate that our test-time on-target adaptation could achieve comparable performance compared to state-of-the-art training-time methods on ImageNet-LT and iNaturalist-18 datasets. We report results for two popular ConvNets, ResNet-50 and ResNet-101, as teacher models trained on long-tailed data. Our method achieves comparable overall performance with the train-time method (LWS) on both

| method | iNaturalist18 | | | | ImageNet-LT | | | |
| --- | --- | --- | --- | --- | --- | --- | --- | --- |
| | many | medium | few | acc. | many | medium | few | acc. |
| ResNet-50 | 72.2 | 63.0 | 57.2 | 61.7 | 64.0 | 33.8 | 5.8 | 41.6 |
| + LWS (Kang et al., 2019) | 65.0 | 66.3 | 65.5 | 65.9 | 57.1 | 45.2 | 29.3 | 47.7 |
| + On-Target Class Distribution | 64.2 | 66.3 | 65.9 | 65.9 | 55.7 | 46.0 | 28.6 | 47.4 |
| ResNet-101 | 75.9 | 66.0 | 59.9 | 64.6 | 66.6 | 36.8 | 7.1 | 44.2 |
| + LWS (Kang et al., 2019) | 69.6 | 69.1 | 67.9 | 68.7 | 60.1 | 47.6 | 31.2 | 50.2 |
| + On-Target Class Distribution | 66.5 | 69.1 | 68.3 | 68.5 | 58.9 | 48.7 | 31.8 | 50.3 |

Table 6: Comparing our method performance with learnable weight scaling (LWS) on long-tailed benchmarks including iNaturalist18 and ImageNet-LT. Note that LWS adapts during training while our method can adapt during testing, which is more efficient.

| network | imagenet pretrain | source only | SHOT | | ours | | network | imagenet pretrain | source only | TENT | | ours | |
| --- | --- | --- | --- | --- | --- | --- | --- | --- | --- | --- | --- | --- | --- |
| | | | avg. | acc. | avg. | acc. | | | | avg. | acc. | avg. | acc. |
| ResNet-50 | ✗ | ✗ | 49.7 | 48.3 | 69.1 | 67.3 | ResNet-50 | ✗ | ✗ | 32.8 | 31.4 | 50.9 | 49.9 |
| ResNet-50 | ✓ | ✗ | 75.0 | 74.5 | 77.8 | 78.6 | ResNet-50 | ✓ | ✗ | 60.9 | 60.2 | 75.1 | 73.8 |
| ResNet-101 | ✓ | ✗ | 82.3 | 77.8 | 85.9 | 82.8 | ResNet-18 | ✗ | ✗ | 34.0 | 33.1 | 51.4 | 51.4 |
| ResNet-101 | ✓ | ✓ | 49.9 | 55.5 | 60.0 | 65.6 | ResNet-50 | ✗ | ✓ | 17.8 | 21.8 | 22.3 | 29.1 |

Table 7: Classification accuracy on VisDA-C (validation). "Imagenet pretrain" indicates whether we utilize ResNet pretrained on ImageNet at stage 0. "Source only" indicates whether we skip test-time adaptation (stage 1) and directly use the source model to generate pseudo labels at stage 3.

datasets. Compared to the vanilla ResNet-50 and ResNet-101, our fully test-time method significantly improves the overall performance by a large margin. Considering the accuracy of few-shot (less than 20 samples) categories, our method outperforms the train-time practice on three out of four cases, extending the usage scenarios of train-time long-tailed recognition methods.

## 4.5 Ablation Studies

**Stage 0: network & initialization** The upper part of Table 7 presents the numbers of SHOT/TENT-IM with ResNet in various depths (18, 50, 101) and initializations (from scratch, ImageNet pretrain). We observe that our method consistently improves the teacher models with various model architectures and initializations, which indicates the usability and versatility of the proposed framework. When adapting from synthetic to real-world domains, the ImageNet pretrained model should not be utilized to start with, due to the learned inductive bias of its parameters. Therefore we also experiment on training from scratch for teacher models. Larger capacity does not lead to better generalization when training from scratch. On the contrary, We observe that a deeper ImageNet pretrained ConvNet provides a stronger inductive bias from the beginning.

**Stage 1: test-time adaptation** The lower part of Table 7 presents the numbers of source-only models as teacher, without any source-free adaptation (TENT-IM/SHOT). The final accuracy after teacher-student suffers from the poorer quality of the initial pseudo label. ImageNet pretraining could alleviate such a phenomenon, but the numbers with test-time adapted teachers are still significantly better than the source-only ones. Comparing these empirical results with table 2, TENT-IM boosts the initial/final accuracy by 15.0/28.6 points, while SHOT brings 32.4/25.9 points improvement. In a word, test-time adaptation should be leveraged in preparation for trustworthy pseudo labels.

**Stage 2: on-target feature** The left part of Table 8 presents the ablation study of student architecture and initialization. When the student model is not initialized on target data, such as source data (VisDA-C train) or the external large dataset (ImageNet), the overall accuracy drops 4+ points, which indicates the necessity of on-target feature learning. We also ablate the same contrastive learning algorithm (MoCo v2) on a different data source, such as VisDA-C val and ImageNet. The empirical results indicate that more external data does not bring any advantage, which echos our statement on the target-specific representation learning.

**Stage 2: contrastive learning** The right part of Table 8 presents the results with various contrastive learning frameworks, including SwAV, SimSiam, Barlow Twins. We train these contrastive learned models with the same number of epochs to have a fair comparison. Compared to the performance of the teacher model, all these learned features bring a noticeable improvement, achieving the com-

| method | avg. | acc. | method | avg. | acc. |
|---|---|---|---|---|---|
| TENT | 32.8 | 31.4 | Ours (MoCo) | 50.9 | 49.9 |
| VisDA-C train | 45.3 | 43.3 | SwAV | 50.5 | 49.4 |
| ImageNet | 47.3 | 46.2 | SimSiam | 48.7 | 47.6 |
| ImageNet (MoCo) | 46.6 | 45.5 | Barlow Twins | 46.3 | 44.8 |

Table 8: Classification accuracy on VisDA-C (validation) . Left side: Ablation results on the student model with various initialization. Right side: Ablation results on the contrastive learning method using MoCo, SwAV, SimSiam, and Barlow Twins.

Figure 5: Example images from VisDA-C, ImageNet, ImageNet-Sketch, and Office-Home.

| teacher | soft | phase 0 | phase 1 | phase 2 | phase 3 | phase 4 | phase 5 | phase 6 | phase 7 | phase 8 | phase 9 |
|---|---|---|---|---|---|---|---|---|---|---|---|
| TENT | ✗ | 32.8 | 44.2 | 48.2 | 50.9 | 52.7 | 54.3 | 56.0 | 57.0 | 58.1 | 58.9 |
| | ✓ | 32.8 | 44.3 | 53.5 | 56.4 | 59.9 | 61.4 | 63.6 | 64.2 | 65.1 | 65.6 |
| SHOT | ✗ | 82.3 | 84.8 | 85.5 | 85.9 | 86.2 | 86.3 | 86.3 | 86.3 | 86.2 | 86.3 |
| | ✓ | 82.3 | 84.7 | 85.2 | 85.6 | 85.2 | 85.7 | 85.0 | 85.4 | 84.9 | 85.2 |

Table 9: Classification accuracy of our method on VisDA-C (validation). "Soft" indicates that the hard-label cross-entropy loss is replaced with the soft-label KL divergence loss for each even-numbered phase. Note that our default number of phases (phase 3) is highlighted.

parable numbers with the reference performance of MoCo v2. We observe that our method is not sensitive to the choice of contrastive learning method. In this way, our on-target adaptation could be further improved by introducing a more advanced contrastive learning approach in the future.

**Stage 3: more phases** Table 9 presents the detailed numbers for each phase during teacher-student. We observe that the first phase already significantly outperforms the test-time adapted teacher model, which is also the state-of-the-art practice. The following several phases gradually improve the results, taking the last generation student as the next teacher. Considering the speed-accuracy trade-off, we choose to have three phases as our default setup, even though more phases could lead to a better result. For example, 9-phase ($3\times$) optimization brings up around 9 points improvement compared to our default 3-phase ($1\times$) one with TENT-IM as the initial teacher model.

**Stage 3: soft label** Table 9 presents the ablation study on the design choice of loss function. Our default setup only utilizes hard labels with cross-entropy loss. Actually, our framework also benefits from the soft label with KullbackLeibler divergence loss, following the popular practice of knowledge distillation (Hinton et al., 2015). We observe that the mix of both hard and soft label bring up the best performance. we replace the cross-entropy loss (hard label) with KullbackLeibler divergence (soft label) for the *even* number of phases. We set the number of epochs as one for all these interpolated soft label phases for a more computational-friendly practice. The known drawback is that the soft label part typically needs specific tuning on learning rate, loss weight, temperature, and so on. Existing works (Berthelot et al., 2019b;a; Sohn et al., 2020) discuss sharpening (temperature) and thresholding (confidence threshold) to improve the performance of semi-supervised learning. Instead, we only ablate the default KullbackLeibler divergence loss without bells and whistles like temperature and confidence threshold. Our default training objective chooses to be the most robust hard label with cross-entropy criterion for all the other experiments.

## 5 CONCLUSION

Domain adaptation is itself adapted to many different needs: unsupervised domain adaptation jointly optimizes over labeled source and unlabeled target data, source-free adaptation adapts to target given source parameters instead of source data, and test-time adaptation even adapts while making predictions. Across each of these varieties, the source comes first. The target representation is either aligned to the source representation or it is initialized from it by transfer learning. On-target adaptation departs from this standard practice by transferring the source predictions without the source representation. This decoupling is unconventional, but useful, because it enables learning all of the target model parameters on the target data. Given enough target data, on-target adaptation improves accuracy by learning the model for target data on target data.

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
