# OpenReview forum: "On-Target Adaptation"
_ICLR.cc/2022/Conference — ICLR 2022 Submitted_

### Official Review · Reviewer_pNRq · 2021-10-26

**Correctness:** 2
**Technical Novelty And Significance:** 3
**Empirical Novelty And Significance:** 2
**Recommendation:** 5
**Confidence:** 4

**Main Review:**

**Strengths:** Interesting and intuitive idea that (i) is well motivated; and (ii) seems to perform well on a variety of datasets.

**Weaknesses:**
  - **Missing critical details:**
    - *Loss function:* It is not clear what is actually being done in stage 3 (teacher-student) due to the lack of an equation for the final loss function, e.g. soft vs. hard losses, "consistency regularization" vs. pseudo-labelling, etc. This should be added to Section 3 to better describe the method.
    - *Algorithm:* It is not clear what is actually being done in stage 3 (teacher-student) due to the lack of an algorithm, e.g. the replacing student and teacher models at certain phases, the swapping between soft and hard losses and certain phases, the re-initializing of student features at certain phases (to correct for incorrect pseudo-labels), etc. This should be added to Section 3 (or Suppl. Material, and pointed to) to better describe the method.
  - **Lack of uniformity in the evaluation (cherry picking?)**:
    - *Table 1:* Why are the 4 entries in the bottom right corner not filled in?
    - *Table 3:* Why does the right column with SHOT not contain the same rows as the other two columns, namely R50S and R50P? Why has R50S been replaced with R101P? Do we not want to see the performance of SHOT with R50S?
    - *Table 4*: Why are SHOT and SHOT + ours omitted from this table?
    - *Table 5*: Why are SHOT and SHOT + ours omitted from this table?
    - *Table 7:* Why have different networks (“network”), pre-training strategies (“imagenet pretrain”) and adaptation strategies (“source only”) been used for SHOT (left) and TENT (right)? Are these the combinations where your method achieved the biggest improvement?
  - **Some prior/related work is not properly cited**:
    - *Not transferring source parameters:* Prior work has already *not* transferred source parameters in the setting of *black-box* source-free domain adaptation [2,3,4]. In fact, Fig. 1c exactly depicts this setting of black-box source-free domain adaptation. This should be discussed.
    - *Factorizing the representation and classifier:* SHOT[1] does exactly this, training the representation while holding fixed the classifier. I think you should cite them in Section 3 (last paragraph of page 3) and perhaps the introduction. Otherwise it (wrongly) comes across as a contribution of this paper.
    - *L_{ent} and L_{div}:* the use of both L_{ent} and L_{div} losses was introduced by SHOT[1] (i.e. their SHOT-IM loss), and should be cited accordingly in Section 3, Stage 1 (teacher).
    - *Using SHOT-IM loss with TENT (Section 4.2, Stage 1):* Since you replace the loss of TENT (L_{ent}) with that of SHOT-IM (L_{ent} and L_{div}), I don’t think you can refer to this method as TENT anymore because it is no longer TENT. Rather, it is a mix between TENT and SHOT-IM and should be labelled as such, e.g. TENT-IM.

**Further comments**:
 - **"Test-time":** I find the use of "test-time" a bit misleading for the *adaptation phase* of source-free domain adaptation. One is not really "testing" at that time, but rather adapting a model to the target data for many epochs (could be several days). Instead, it could simply be called adaptation, target adaptation, or on-target adaptation! The addition of “offline” and “online” test-time approaches further added to my confusion.
 - **"Label smoothing *could* be leveraged":** Was label smoothing used or not? (Section 4.2, Stage 0).


[1] Liang, J., Hu, D., & Feng, J. (2020). Do we really need to access the source data? Source hypothesis transfer for unsupervised domain adaptation. In *International Conference on Machine Learning* (pp. 6028-6039).

[2] Liang, J., Hu, D., He, R., & Feng, J. (2021). Distill and Fine-tune: Effective Adaptation from a Black-box Source Model. *arXiv preprint arXiv:2104.01539*.

[3] Zhang, H., Zhang, Y., Jia, K., & Zhang, L. (2021). Unsupervised Domain Adaptation of Black-Box Source Models. *arXiv preprint arXiv:2101.02839*

[4] Wu, K., Shi, Y., Han, Y., Shao, Y., & Li, B. (2021). Black-box Probe for Unsupervised Domain Adaptation without Model Transferring. *arXiv preprint arXiv:2107.10174*.


**Summary Of The Paper:**

This work addresses the problem of source-free domain adaptation. Instead of fine-tuning *the source model* on the target data, this work proposes to fine-tune *a representation learned on the target data alone* using e.g. self-supervised contrastive learning. This permits distinct target architectures that can improve performance and/or reduce memory and computational cost.

**Summary Of The Review:**

Overall, this seems to be a good paper. A well-motivated and intuitive idea is proposed, and it appears to work well across a variety of datasets. However, critical details are omitted (e.g. what is the complete loss function and algorithm?), there is a lack of uniformity in the evaluation (raising suspicions of cherry picking), and related works are not cited satisfactorily (e.g. black-box source-free domain adaptation). As a result, I believe this paper is only marginally above the acceptance threshold in its current form. If the aforementioned concerns can be addressed, I will improve my score.

---

> ### Author Response · Authors · 2021-11-23
> **Revision and Related Concurrent Work**
>
> Thank you for your feedback. We appreciate the detailed suggestions and your providing pointers to exciting concurrent work on arxiv!
>
> > Missing critical details
>
> We have included an algorithm description as pseudo-code, edited table captions, and edited the text about label smoothing.
>
> > Lack of uniformity in the evaluation
>
> Please see "Evaluation" under our general response. Our experiments report both controlled comparisons and exploratory results where we use our new capacity to change the target architecture. The reported combinations actually favor the compared methods, as we clarify in using the preferred configuration of SHOT for instance. We have also included R50S in Table 2 as requested.
>
> > Related Work
>
> Thank you for these exciting references on black-box adaptation! Please note that these are concurrent works which are on arxiv but not already published. As they are indeed related, we now include these references in Section 2 of the revision. Please "Novelty" in our general response for a contrast between our work and theirs, which is summarized by our focus on self-supervision and the target representation, rather than the teacher and student predictions and regularization.

---

> > ### Comment · Reviewer_pNRq · 2021-11-27
> > **Response to authors**
> >
> > Thank you for your response. I appreciate the algorithm being added (although it still lacks a lot of the detail I previously mentioned) and black-box SFDA being mentioned. However, many of my concerns were either ignored or not satisfactorily addressed. In particular, my concern over the fairness of the evaluation has grown, since many of the points raised were not addressed satisfactorily. As a result, I lower my score to 5.

---

### Official Review · Reviewer_QwsH · 2021-10-31

**Correctness:** 3
**Technical Novelty And Significance:** 3
**Empirical Novelty And Significance:** 3
**Recommendation:** 6
**Confidence:** 3

**Main Review:**

Pros:
* I like the source-free adaptation problem setting, I think it is very practical
* Combining model fine-tuning on target (or invoking a SOTA source free method) generating a teacher for distillation and using contrastive self-supervised student is a natural idea to try and I think the community should be aware of the good results
* SOTA improvements in some cases are welcomed
* extensive experiments and ablations
* code included

Cons:
* novelty seems a bit limited, the paper combines some standard ideas (target adaptation, distillation, contrastive pre-train on target, fix-match), the main novelty seems to be in the system - combining these for source free adaptation
* writing could be improved, sometimes it is hard to follow the mass of details and some conclusions become less clear
* why in Table 1 some numbers are missing and X is drawn? did not find an explanation in the text
* in Table 2 "Ours" is only shown on top of TENT or SHOT, how about without source free sota methods at play? is it in table 3? but there the numbers are a bit low and disappointing...
* Tables 4 & 5 only do TENT / TENT + ours, I looked briefly in SHOT and saw higher numbers for Table 5 as it seems, though there the proposed method did not hit sota anyway, how about adding SHOT also to table 4 (and 5)?
* Presentation (more minor): ablation tables are better re-designed, e.g. look at table 7 the message there was to show that there is improvement in every row reading from left to right, but the V and the X leads the reader to wonder if adding more things improves or not in columns and it is actually opposite in that table (but was so in earlier tables). In short - please make the style consistent, it is really hard to follow. This was just an example.

**Summary Of The Paper:**

The paper proposes a multi-stage approach for source-free domain adaptation, namely when source data is not available and one can only use the model pre-trained on the source and adapt it based on unlabelled target domain data. The paper advocates for learning on target not via fine-tuning the source model, but rather distill from it initializing on source using contrastive learning (the authors use Moco V2). The source model is also adapted on target via InfoMax loss before being used as the teacher for the distillation. The authors propose to use the FixMatch strategy (with strong and weak aug taken from AutoAugment) for the distillation. The authors show some gains when combining with SOTA source-free adaptation methods: TENT and SHOT. The authors also show that improved accuracy can be obtained with much smaller models trained on target using their method (e.g. replacing Res50 or Res101 with Res18 on Visda-C. Extensive experiments, ablations, and results are provided.

**Summary Of The Review:**

I see a balance between pros and cons in this paper, yet I do have some positive tendency as the method itself is clear and intuitive to me and can boost performance in some interesting cases (not always, but still). Therefore, at least until I see some more clarifications from the authors and other reviews, I prefer to rank it slightly above borderline, with an intention to revisit and yet in a positive spirit of leaning towards acceptance :-)

---

> ### Author Response · Authors · 2021-11-23
> **Novelty and Table Clarifications**
>
> Thank you for your feedback. We appreciate your recognition of the source-free adaptation setting, the interest in self-supervision for this purpose, and the improvement possible with our approach. We appreciate all the more the constructive and concise points to address:
>
> > the main novelty seems to be in the system - combining these for source free adaptation
>
> We agree, and underline in our general response (Novelty) that the novelty is one part conceptual—in how a system can learn more on target—and one part empirical—in reporting results that show the system can improve both accuracy and computational efficiency by contrastive learning and altering the target architecture. While some parts of our system are standard, its overal characteristic in training 100% of the target model parameters on target data is not (as illustrated in Figure 1).
>
> > why in Table 1 some numbers are missing and X is drawn?
>
> We have revised Table 1 to focus on each stage and the contribution of our on-target representation from contrastive learning. The results with X did not include contrastive learning, as the state-of-the-art contrastive methods we experiment with had not yet been fully explored on these datasets.
>
> > in Table 2 "Ours" is only shown on top of TENT or SHOT, how about without source free sota methods at play?
>
> Please note that this comparison is shown in Table 1 for ViSDA-C, a gold standard benchmark dataset. On-target representation learning and teacher-student updates help with or without the source-free adaptation methods.
>
> > how about adding SHOT also to table 4 (and 5)?
>
> Thank you for this suggestion. For time and space we could only report TENT(-IM) for these tables, but we include comparison with SHOT in Table 2 and 3. We will add SHOT results in the supplement for these tables in the next version.

---

> > ### Comment · Reviewer_QwsH · 2021-12-01
> > **in response to authors clarifications**
> >
> > Thank you for your response. I see that some of my concerns were addressed, but for some, I did not find a satisfactory response. E.g. on missing results in the tables or not including SHOT. However, I still believe that the paper is worth publishing and prefer to keep my original rating of "marginally above ...".

---

### Official Review · Reviewer_Lx65 · 2021-11-06

**Correctness:** 3
**Technical Novelty And Significance:** 3
**Empirical Novelty And Significance:** 3
**Recommendation:** 3
**Confidence:** 3

**Main Review:**

Strength:  The paper discusses a direct target adaptation to improve the performance of target data ( which is the partial objective of the traditional domain adaptation method). The paper shows a different variation of each component ( initialization, self-supervised, etc.). The paper is very easy to follow.

Weakness: The paper needs to improve significantly in terms of proper evaluations ( fair evaluations). Some of the points are listed here:
 The paper's novelty is limited; it uses most of the well-studied techniques such as Infomax-based entropy loss, self-supervised methods, and distillation. The paper seems it is only combinations of those methods without proper justification ( motivation) why should these particular techniques are suitable for the current problem. Although they showed it only based on empirical performance, it should not be the motivation or intuition to use particular techniques.
In stage 1, the parameters of the feature extractor are updated, not the classifier; I wonder what would be effective if we update the classifier too. This will help the feature to be class discriminative for the target domain rather than the source domain.
Regarding comparing state-of-the-art methods, the performance of domain adaption methods on the source domain does not degrade while adapting the target domain. In this case, we concern only with target performance, so I suspect the model will perform poorly on the source domain. So the comparison in terms of the target domain is not fair enough.
In Table 2, the performance reported on TENT ( baseline)  with network  R50S but the proposed model trained on R18S; what is the reason to choose different base networks. For the fair comparing the base network must be the same for all the experiments. The same thing for the SHOT baseline.
In Table 5, again, the base network is different from the baseline or other state-of-art methods. In fact, authors should include the source-only models' performance in each table, which is the super baseline for all the source-free or ( not source-free) domain adaption methods.
Overall due to the missing proper fair comparison, the results do not show the effectiveness of the proposed methods.
Thus I would like to see the results with fair base network.



**Summary Of The Paper:**

The paper proposes a direct target adaptation from the source trained model (without the source data).  It applied self-supervised and distillation methods to learn from unlabeled target data.

**Summary Of The Review:**

The paper needs a proper evaluation on the fair baseline. The weaknesses are discussed above.

---

> ### Author Response · Authors · 2021-11-23
> **Target Accuracy is the Purpose of Adaptation and Fair Evaluations are Included**
>
> Thank you for your feedback. We appreciate the importance of evaluation, and so address this topic in our general response (Evaluation) and in this response.
>
> > the performance of target data ( which is the partial objective of the traditional domain adaptation method)
> > the comparison in terms of the target domain is not fair enough
>
> We respectfully disagree, because target accuracy is a principal goal of domain adaptation [Quionero-Candela et al. MIT Press '09, Saenko et al. ECCV'10, and the references of Section 2]. Our experiments therefore measure target accuracy, not source accuracy, since source accuracy is the focus of supervised learning in general.
>
> > authors should include the source-only models' performance in each table, which is the super baseline for all the source-free or ( not source-free) domain adaption methods
>
> The reported adaptation methods are more accurate than the source-only model, which is a weak baseline in each case, so we report only the competitive adaptation methods for space. (See Table 1 for an example of poor source-only accuracy.)
>
> > the base network must be the same for all the experiments
>
> Please note that a contribution of our work is that the target model architecture does not need to be the same as the source architecture. This enables choosing a smaller target architecture for smaller target datasets.
>
> At the same time, we fully agree it is important to measure this new capacity by controlled experiments. Please see Tables 4 & 7 of the original submission for such comparisons with the same base architecture, and the newly included R50S result in Table 2 of the revision. In each case our on-target adaptation still improves.
>
> > In stage 1, the parameters of the feature extractor are updated, not the classifier; I wonder what would be effective if we update the classifier too
>
> Please see the discussion of adaptation parameters in the cited TENT and SHOT papers. Both find that it is more effective to adapt a subset of parameters than to directly optimize the classifier parameters during this stage. Note that the design choices of Stage 1, for source-free adaptation, are not the subject of our contribution. We investigate if parameters can be learned fully on target data by self-supervision on target and distillation from source predictions, where those source predictions can be improved by Stage 1. Our on-target adaptation differs in its Stage 2, with contrastive learning, and its training of the classifier parameters on target data via distillation and teacher-student updates with Stage 3.
>
> We hope that these comments along with our general response regarding novelty and evaluation have clarified the reason for reporting the experiments we have with different representations and different architectures.

---

### Author Response · Authors · 2021-11-23
**General Response for All Reviewers**

We thank the reviewers for their attention and feedback! We reply with general responses to shared topics across reviews, and note that we have posted a revision of the paper that incorporates several points of feedback.

**Purpose**
The goal of this work is to improve accuracy on the target data by optimizing more on the target data. Our work brings up the possibility of fully learning the target model on target data by using the supervision of source predictions without the transfer of the source representation. Our work is the first to show that self-supervision can serve this purpose by initializing the target representation with contrastive learning instead of transferring the source representation.

**Novelty**
The contributions are not in any one technical detail, but in our conceptual insight that _100% of model parameters_ can be optimized on target data and our empirical demonstration that doing so by _self-supervision_ and distillation can improve task accuracy and computational efficiency.

- Our on-target adaptation approach optimizes 100% of the target model parameters on target data, unlike prior published adaptation methods for unsupervised domain adaptation and source-free adaptation. Existing approaches rely on either joint optimization on source and target or transfer from source to target. Closely-related source-free methods do without source data, but nevertheless transfer source parameters.
- Our on-target representation is optimized on the target data by self-supervision with contrastive learning, which is a loss on the input data. This stands in contrast to an exciting wave of concurrent works on "black-box" adaptation that are supervised entirely by losses and regularizers on the output like distillation [2, 3, 4 in review pNRq]. Our experiments are the first to show the suitability of self-supervised initialization for source-free adaptation, and is therefore complementary to the concurrent work that focuses on improving distillation and teacher-student learning for adaptation, which is likewise an important direction. We are encouraged to see our on-target and their black-box methods both argue for adaptation from source predictions rather than source parameters.

**Evaluation**
Our experiments report common metrics on standard benchmarks with fair comparisons. When comparing to other methods, we choose configurations that either control for differences (like architecture, etc.) or respect the best settings reported by the original works. Please note that our tables have controlled results, where architectures are held the same, plus additional results that compare to best settings (for example Table 3's ResNet-101-P for SHOT, which is its preferred configuration in [Liang et al. ICML'20]) or vary an option for our method like architecture.

Along with our response we have uploaded a revision to the paper to clarify points raised by review, including:

The algorithm is summarized as pseudocode by FIgure 4.
Table 2 now includes a result with R50S as an additional control for architecture.
The concurrent work from arxiv (see review pNRq) is discussed in Section 2.
Label smoothing is not used except for SHOT as specified in Section 4.

---

### Decision · Program_Chairs · 2022-01-20

**Decision:**

Reject

**Comment:**

The work proposed an interesting source free adaptation setting, where one is asked to adapt a pre-trained source model to a target domain without accessing data from the source domain. While reviewers find the setup interesting and the initial results encouraging, they expressed concerns on the limited novelty of the work as well as incomplete evaluation. Multiple reviewers (reviewer Lx65  and pNRq) raised concerns on the fairness of the evaluation, which was not fully addressed by the authors during rebuttal. Please consider addressing these comments in your draft.